# *Lacticaseibacillus rhamnosus* Probio-M9 Alters the Gut Microbiota and Mitigates Pulmonary Hypertension in a Rat Model

**DOI:** 10.3390/nu17182927

**Published:** 2025-09-11

**Authors:** Zhixin Zhao, Gaopeng Li, Kiyomi Ohmichi, Xiaodong Li, Feiyan Zhao, Kaori Ishikawa, Ryou Ishikawa, Kazufumi Nakamura, Naoya Yokota, Zhihong Sun, Lin Hai Kurahara

**Affiliations:** 1Inner Mongolia Key Laboratory of Dairy Biotechnology and Engineering, Key Laboratory of Dairy Products Processing, Ministry of Agriculture and Rural Affairs, Key Laboratory of Dairy Biotechnology and Engineering, Ministry of Education, Inner Mongolia Agricultural University, Hohhot 010018, China; 2Department of Cardiovascular Physiology, Faculty of Medicine, Kagawa University, Takamatsu 761-0793, Kagawa, Japan; 3Department of Diagnostic Pathology, Kagawa University Hospital, Takamatsu 761-0793, Kagawa, Japan; 4Department of General Medicine, Kagawa University Hospital, Takamatsu 761-0793, Kagawa, Japan; 5Center for Advanced Heart Failure, Okayama University Hospital, Okayama 700-8558, Okayama, Japan; 6Department of General Thoracic Surgery, Faculty of Medicine, Kagawa University, Kita-gun, Miki-cho 761-0793, Kagawa, Japan

**Keywords:** pulmonary artery remodeling, probiotics, gut microbiota, macrophages, GPNMB, CD44

## Abstract

**Background**: Intestinal microbiota plays an important role in the progression of pulmonary hypertension (PH). Colostrum-derived *Lacticaseibacillus rhamnosus* Probio-M9 (Probio-M9) has shown protective effects against inflammation and remodeling. We investigated whether Probio-M9 supplementation could improve the pathology of PH. **Methods**: The monocrotaline (MCT)-induced PH model rats are created followed by Probio-M9 treatment. Microbiota and pathological analyses were performed to investigate the therapeutic effects of Probio-M9. **Results**: Probio-M9 significantly suppressed cardiovascular remodeling and reduced mortality in rats. Analysis of the fecal microbiota revealed that Probio-M9 significantly altered the gut microbiota of MCT model rats. Specifically, *Alistipes* sp009774895 and *Duncaniella muris* populations increased, whereas *Limosilactobacillus reuteri*_D, *Ligilactobacillus apodeme* and *Monoglobus* sp900542675 decreased compared to those in the MCT group. Focusing on the expression of GPNMB in macrophages and the localization of CD44, we found that the number of these cells increased in the MCT group but significantly decreased with Probio-M9 treatment. In lung tissue from PH patients, more GPNMB-positive macrophages were found than non-PH lungs, and an increase in CD44-positive cells was confirmed in the vicinity of GPNMB. **Conclusions**: Probio-M9 had a significant impact on the intestinal microbiota and GPNMB/CD44 positive cells in the lungs of PH rats.

## 1. Introduction

Pulmonary hypertension (PH) is a serious cardiovascular disease characterized by pulmonary artery (PA) remodeling. Referring to European data, PH was estimated at 5.8 cases per million adults annually, while its prevalence ranges from 47.6 to 54.7 cases per million adults [1]. Current clinical treatments are limited to the use of vasodilators, which have limited therapeutic efficacy. Numerous studies have reported changes in the intestinal flora associated with PH, and intestinal microbiota transplantation has been performed [2,3]. We investigated the therapeutic effects of probiotics in a rat model of PH. *Lacticaseibacillus rhamnosus* Probio-M9 (Probio-M9), a probiotic isolated from human colostrum, has been reported to regulate intestinal flora, exert anticancer effects, and modulate immune checkpoints involving T cells, thereby strengthening the immune system and potentially extending the lifespan [4,5,6,7].

In the lungs affected by PH, there is not only abnormal proliferation of pulmonary artery smooth muscle cells (PASMCs), but also an accumulation of immune cells such as macrophages, which have garnered attention, particularly those expressing glycoprotein non-metastatic B (GPNMB) [8]. The single-cell RNA sequencing on NCBI Gene Expression Omnibus databases, analyzed lung tissue samples across healthy controls and Pulmonary arterial hypertension (PAH) patients to find the significant changes and potential functions of myeloid cell subsets in PAH patients and especially focused on GPNMB+ macrophages [8]. GPNMB is a cell membrane glycoprotein mainly involved in cell–cell interactions and cell proliferation [9]. GPNMB plays a role in tissue homeostasis by interacting with the extracellular matrix [10]. It is particularly expressed in macrophages and is implicated in the abnormal proliferation and remodeling of lesions. CD44 has been identified as a GPNMB receptor and is expressed in cancer stem cells and T cells in the lungs [11,12]. The localization of GPNMB+ macrophages and CD44+/αSMA+ cells have been identified in the mice PH model [8].

The lung–gut axis refers to the interactions and relationships that exist between the lungs and intestines [13,14]. This correlation indicates that the lungs and intestines mutually influence each other through the immune, nervous, and hormonal systems [15]. First, the microbial community in the intestine (gut flora) influences the immune system throughout the body, impacting immune responses in the lungs. Changes in the gut bacteria can affect lung diseases [16]. Second, stress and emotional states can affect gut and lung function through the vagus nerve, which connects the brain and abdomen [17]. Third, various inflammatory cytokines secreted from the intestine regulate the immune response in the body, thereby affecting the lungs [18]. It has been suggested that adjusting the gut flora may also contribute to lung health and that bacterial supplements can ameliorate or induce PH, affect pulmonary vascular remodeling mechanistically (e.g., via systemic inflammation, metabolite production such as SCFAs, or immune cell trafficking) [19,20,21,22].

Although the role of gut microbiota in PH has been explored, no studies to date have examined the effects of Probio-M9 on both gut microbiota composition and GPNMB/CD44 expression in PH. We hypothesized that Probio-M9 supplementation would remodel the gut microbiota and modulate immune responses in the lungs, thereby reducing vascular remodeling and improving outcomes in a monocrotaline (MCT)-induced PH rat model. In addition to analyzing gut bacteria, we also investigated the changes in the expression of GPNMB and CD44 in the lungs. Understanding the interactions between the lung and intestinal microbiota may provide valuable insights into novel therapeutic approaches for PH.

## 2. Materials and Methods

### 2.1. Monocrotaline (MCT)-Induced PH Model Rats

All animal experiments were conducted in accordance with the institutional and National Institutes of Health guidelines for the care and use of laboratory animals. Rats were housed at 23–25 °C in a humidity-controlled colony room, maintained on a 12 h light/dark cycle, and provided with a standard diet and water. Eight-week-old male Sprague-Dawley rats received a subcutaneous injection of monocrotaline (MCT, Sigma-Aldrich Chemical Co., St. Louis, MO, USA) (60 mg/kg) on day 1 (Figure 1a). The experimental groups were divided into control group (CTR), MCT group (MCT), and MCT + Probio-M9 (MCT + M9) group. Probio-M9 (4 × 10^9^ CFU/day) was administered orally in drinking water starting 10 days after MCT injection. Probio-M9 administration does not change the water and food intake of the rats. Two sets of experiments were conducted: one for the evaluation of cardiac function on day 18 and survival until day 32 (*n* = 10), and the other for the collection of tissues and stools for evaluation on day 23 (*n* = 8). Randomization and blinding were applied for all experiments.

### 2.2. Echocardiography in Rats

Anesthesia was induced in rats using 5% isoflurane and maintained using 2% isoflurane (Fujifilm-wako, Osaka, Japan). Echocardiography was performed randomly in a blinded manner using a LOGIQ ultrasound machine (GE Healthcare, Chicago, IL, USA) and equipped with a 5–11.5 MHz multifrequency probe. The probe was gently applied to the left thorax and adjusted to obtain optimal images of the ventricle and PA. Right ventricular (RV) systolic and diastolic functions and ventricular wall thickness were evaluated. Once the procedure was completed, anesthesia was discontinued, and the rats were placed in a warm recovery cage and monitored until fully awake. The recovery status of the rats was monitored and documented to ensure their wellbeing. Subsequent procedures, including tissue sampling, were performed in accordance with the ethical guidelines for animal experiments.

### 2.3. Immunofluorescence Staining of the Lung

Formalin-fixed paraffin-embedded tissues were deparaffinized, and antigen retrieval was performed by incubating them in citrate buffer (pH 6.0) at 95 °C for 20 min. After blocking with Blocking One Histo (Nacalai Tesque, Kyoto, Japan), the sections were incubated overnight at 4 °C with primary antibodies. The antibodies used are listed in Table 1.

Subsequently, the sections were incubated with fluorescent dye-conjugated secondary antibodies for 60 min at room temperature. Nuclear staining was performed using DAPI (4′,6-diamidino-2-phenylindole) (Invitrogen, Thermo Fisher Scientific, Waltham, MA, USA). Fluorescence images were obtained using a confocal laser-scanning fluorescence microscope (LSM 700; Carl Zeiss, Oberkochen, Germany).

### 2.4. Clinical Samples for Histological Evaluations

Formalin-fixed, paraffin-embedded lung tissue samples were used in this study. Lung tissues of patients with idiopathic pulmonary arterial hypertension (IPAH) and those without PH or PA remodeling were obtained by biopsy and autopsy at Okayama University and Kagawa University (Table 2).

### 2.5. Extraction of Fecal DNA and Metagenomic Sequencing

Fifteen fecal samples were collected from each group (CTR = 5, MCT = 5, and MCT-M9 = 5) after the intervention. Metagenomic DNA was extracted from fecal samples using the QIAamp Fast DNA Stool Mini Kit (Qiagen GmbH, Hilden, Germany) following the manufacturer’s protocol. The extracted DNA’s quality, purity, and integrity were evaluated with a Nanodrop spectrophotometer, 1.0% agarose gel electrophoresis, and the Qubit^®^ dsDNA Assay Kit alongside a Qubit^®^ 2.0 fluorometer (Life Technologies, Carlsbad, CA, USA). Only those DNA samples demonstrating high-quality—concentrations above 20 ng/μL and an OD260/280 ratio ranging from 1.8 to 2.0—were chosen for subsequent processing with the NEBNext^®^ Ultra™ DNA Library Prep Kit for Illumina (New England Biolabs, Ipswich, MA, USA).

### 2.6. Quality Control of Metagenomic Data

A total of 15 fecal samples underwent shotgun sequencing, yielding 104.51 Gb of high-quality paired-end reads, with an average of 6.97 ± 0.79 Gb per sample (ranging from 5.73 to 8.71 Gb). To filter out low-quality sequences and mouse-contaminating reads, KneadData (http://huttenhower.sph.harvard.edu/kneaddata, accessed on 15 February 2025), Trimmomatic (v0.33), and Bowtie2 (v2.5.0) were employed. In the end, 102.75 Gb of clean data were retained for further analysis, resulting in an average of 6.85 ± 0.78 Gb per fecal sample (with a range of 5.63 to 8.57 Gb).

### 2.7. Metagenomic Assembly, Contig Binning, Genome Dereplication

To derive species-level genome bins (SGBs) from the post-quality control metagenomic dataset, MEGAHIT (v1.2.9) was used to assemble the reads from each sample into contigs. Subsequently, MetaBAT2 (v2.15) was applied to the bin contigs longer than 200 bp, resulting in the generation of metagenome-assembled genomes (MAGs). Subsequently, reads were aligned back to the respective contigs using BWA-MEM2 (v2.2.1), with the read depth calculated using SAMtools (v1.18) and the jgi_summarize_bam_contig_ depth function in MetaBAT2. The completeness can be estimated by detecting the proportion of conserved single-copy genes in the MAG. The contamination is calculated by counting the frequency of the occurrence of multiple copies of the same marker gene. The completeness and contamination of the MAGs were evaluated using CheckM (v1.2.2), categorizing them as follows: high-quality (completeness ≥ 80%, contamination ≤ 5%), partial quality (completeness ≥ 50%, contamination ≤ 5%), and medium quality (completeness ≥ 70%, contamination ≤ 10%). Finally, dRep (v3.0.1) was used to cluster and extract SGBs from high-quality genomes, employing the parameter settings -pa 0.95 and -sa 0.95. We ultimately identified 89 SGBs.

### 2.8. Taxonomic Annotation, Abundance, Prediction of Gut Metabolic Modules of SGBs

To obtain species annotations, SGBs were annotated using Kraken2 (v2.1.3) in conjunction with the NCBI (National Center for Biotechnology Information) non-redundant Nucleotide Sequence Database using the default settings. Putative genes were identified using the UniProt Knowledgebase (UniProtKB, release 2020.11) via the BLASTP function of DIAMOND (v2.1.7), with default parameters. Abundance for each SGB was calculated using CoverM (https://github.com/wwood/CoverM, accessed on 7 March 2025) with the parameters “–min-read-percent-identity 0.95–min-covered-fraction 0.4.” Gene abundance was represented as Reads Per Kilobase per million mapped reads (RPKM). Additionally, the predicted open reading frames for each SGB were compared to the Kyoto Encyclopedia of Genes and Genomes (KEGG) database to predict the metabolic modules.

### 2.9. Statistical Analyses

The *n* indicates the number of independent experiments or the number of animals. The sample size *n* = 10 for initial survival study was determined by power analysis. Differences in numerical variables among groups were evaluated using analysis of variance, followed by the Tukey–Kramer test for multiple comparisons. Statistical significance was set at *p* < 0.05. All statistical analyses were performed using Prism-GraphPad ver9.5.1 (GraphPad Software, San Diego, CA, USA).

Statistical analyses of the metagenome analyses were conducted using R software (v. 4.2.1), with data expressed as mean ± SD. The Shannon–Wiener index and Chao 1 richness were calculated using R packages, including vegan, ggpubr, and dplyr, to assess changes in microbial diversity within the fecal samples. Principal coordinate analysis (PCoA) based on the Bray–Curtis distance was employed to evaluate the shifts in microbial structure among the fecal samples. Differences in species and metabolic modules between groups were assessed using the Wilcoxon test, with the significance threshold set at *p* < 0.05. All graphical presentations were created using R and Adobe Illustrator (v28.5).

## 3. Results

### 3.1. Probio-M9 Prolonged Survival and Mitigated the Pathology in MCT Rats

MCT model rats were used to examine the therapeutic effects of Probio-M9. Treatment with Probio-M9 (4 × 10^9^ /day) supplementation starting 10 days after MCT injection significantly suppressed cardiovascular remodeling and prolonged the survival of rats with MCT-induced PH (Figure 1). Echocardiography on day 18 showed that Probio-M9 treatment significantly improved the decreased PA internal diameter (PAID), and right ventricular wall thickness (RVWT) compared with the MCT group (Figure 1b,c). Histopathology on day 23 showed that Probio-M9 treatment significantly decreased the PA wall and RV thickness compared to the MCT group (Figure 1e–g).

### 3.2. Probio-M9 Supplementation Significantly Modulated Gut Microbiota of MCT Model Rat

The fecal microbiota of 15 rats was analyzed using metagenomic sequencing, and 89 SGBs were identified within the complete dataset. There were no significant intergroup differences in the diversity (Shannon–Wiener diversity index and Chao1 richness, Figure 2a,b) and structure (PCoA based on Bray–Curtis distance) of the gut microbiota (Figure 2c). However, in the gut microbiota at a finer level, the fecal microbiota of the MCT group had significantly less Bacteroides sp002491635, Duncaniella muris, Prevotella sp002933775 and Ruminococcaceae sp. compared to the CTR group (*p* < 0.05). The fecal microbiota of the MCT-M9 group contained significantly more Alistipes sp009774895 and Duncaniella muris than that of the MCT group (*p* < 0.05), whereas Limosilactobacillus reuteri_D, Ligilactobacillus apodeme and Monoglobus sp900542675 exhibited the opposite trend (*p* < 0.05). In addition, significantly more Intestinimonas sp900540545 were detected in the MCT-M9 group than in the CTR group (*p* < 0.05; Figure 3). Collectively, these results suggest that Probio-M9 supplementation significantly modulated the gut microbiota of MCT model rats (*p* < 0.05).

### 3.3. Probio-M9 Supplementation Significantly Modulated the Metabolic Processes of Predicted Gut Bioactive Metabolites of MCT Model Rats

We next assessed whether these compositional shifts were associated with functional metabolic changes. To identify the intervention-associated changes in gut metabolic modules, we established a genome-centric metabolic reconstruction that utilized a complete dataset of 89 SGBs, referencing the MetaCyc and KEGG databases. Twenty-four significantly differential metabolic modules were identified in 12 significantly different species among the intergroup comparisons, including glucose metabolism, amino acid metabolism, energy metabolism, short-chain fatty acid (SCFAs) metabolism, P-cresol synthesis, urea degradation, mucin degradation and 17-beta-Estradiol degradation. Notably, Monoglobus sp900542675 is involved in regulating nitric oxide degradation, *Duncaniella muris* and *Ruminococcaceae* sp. are involved in energy metabolism, and *Bacteroides* sp002491635 and *Alistipes* sp009774895 are involved in regulating the synthesis of SCFAs and P-cresol (Figure 4).

### 3.4. Probio-M9 Downregulated Upregulated CD44 and GPNMB Expressing PA in MCT Rats

Next, we focused on macrophage GPNMB expression in the lungs and the localization of CD44, the GPNMB receptor. We found that the number of CD44 positive cells (CTR 4 vs. MCT 12 vs. MCT + M9 7 in median value) and GPNMB-positive macrophages (CTR 7 vs. MCT 19 vs. MCT + M9 10.5 in median value) in the PA region was significantly increased in the MCT group compared to the CTR group, whereas Probio-M9 treatment significantly reduced it (Figure 5a–d).

In the MCT rat lung, remodeling PAs were surrounded by numerous GPNMB+ or CD44+ cells. To evaluate the clinical evidence about these changes, we next analyzed the GPNMB+ or CD44+ cells localization and expression patterns clinical samples.

We also examined GPNMB and CD44 expression in the lungs of patients with and without PH and found that the number of CD44 positive cells (nonPH 12 vs. PH 17 in median value) and GPNMB-positive macrophages (nonPH 5 vs. PH 15 in median value) in the PA region was significantly higher in the PH group than in the non-PH group (Figure 6). The enlarged image in the right panel shows that CD44-positive cells were localized near the GPNMB-positive cells (Figure 6a).

## 4. Discussion

This study is the first to investigate the effects of Probio-M9 on both gut microbiota composition and GPNMB/CD44 expression in PH. PH is a complex and fatal disease characterized by an abnormally elevated PA pressure, which may ultimately lead to right heart failure [23]. In recent years, an increasing number of studies have gradually recognized the important link between PH and the gut microbiota [24], and the association between the gut microbiota and its pathogenesis has attracted widespread attention [25]. Our study preliminarily suggests a potential benefit of Probio-M9 supplementation in alleviating PH. The key findings reveal a significant increase in GPNMB+ macrophages in the lungs of MCT rats, while a marked reduction in these pro-inflammatory macrophages following Probio-M9 treatment. Based on a macrogenomic strategy, our results suggest that the potential mechanism by which Probio-M9 supplementation alleviates PH is through the modulation of beneficial gut microbiota, such as *Alistipes* sp009774895 and *Intestinimonas* sp900540545. This mechanism also involves regulating the metabolic processes of nitric oxide, p-cresol, and butyric acid during the course of treatment.

Our results showed that no significant differences were found in the diversity of the gut microbiota of MCT rats compared to that of CTR rats. This result was consistent with the findings of Hong et al. [26]. Although no effect of PH on the overall microbiome (alpha and beta diversity) was found, this finding further reflects the role of the gut microbiome in different physiological and pathological states. Although MCT rats may exhibit differences in metabolic or other physiological traits, the similarity in the gut microbiota suggests that the diversity of gut microbes may not be the only factor influencing the model. In addition, Probio-M9 supplementation did not significantly alter the diversity of the gut microbiota in MCT rats, which triggered a deeper consideration of the mechanisms of the gut microbiome response. Although Probio-M9 probiotics may show good regulation in other models [7,27,28], in MCT rats, the stability of the gut microbiota in MCT rats may limit changes in its diversity. We speculate that the mechanism by which Probio-M9 alleviates PH may be because of the regulation of specific species or functions. Gut microbiota-derived metabolites may influence vascular remodeling, oxidative stress, and right ventricular function, establishing a stronger link with PH pathophysiology.

Furthermore, Probio-M9 supplementation significantly increased the relative abundance of *Alistipes* sp009774895. *Alistipes* sp. are important beneficial gut bacteria that may play a key role in alleviating PH. *Alistipes* sp. is the main short-chain fatty acid-producing bacterium in the gut and is capable of producing SCFAs, such as propionic acid and butyric acid [29,30]. SCFAs have been shown to suppress pro-inflammatory macrophage activation via histone deacetylase inhibition and G-protein-coupled receptor signaling. Given that GPNMB+ macrophages are known to drive vascular remodeling, their reduction in M9-treated rat may reflect SCFA-mediated immunomodulation [9,12,31]. SCFAs also play an important role in mucosal immunity. SCFAs promote gut immune cell differentiation and mucin production, increase intestinal IgA secretion, and contribute to the maintenance of intestinal barrier function [32]. This protective effect may help reduce the inflammatory response in the lungs and slow the development of PH [33,34]. The decrease in Monoglobus sp900542675 (implicated in NO degradation) suggests improved endothelial function and gut barrier integrity, potentially reducing pulmonary inflammation [35]. This aligns with previous evidence that gut-derived metabolites influence macrophage activation and polarization in distant organs, including the lungs.

In terms of metabolism, Probio-M9 supplementation significantly affected the metabolic processes of nitric oxide, p-cresol and butyric acid. The enrichment of microbial metabolic pathways related to SCFA synthesis and the regulation of P-cresol-producing bacteria in Probio-M9-treated rats may further explain the attenuated pulmonary GPNMB+ macrophages abnormal activation. Changes in the levels of nitric oxide, an important endogenous vasodilator, directly affect the PA pressure [36]. Probio-M9 supplementation may affect the metabolism of nitric oxide by modulating certain gut bacteria (Monoglobus sp900542675), which in turn improves PH. In addition, p-cresol metabolism is associated with oxidative stress and inflammatory responses. Probio-M9 supplementation indirectly affects these metabolic processes and may help mitigate disease progression in PH.

Our results suggest that modulation of specific gut microbial taxa and their metabolites can attenuate vascular remodeling in PH through immunomodulation of lung macrophages. We observed a significant increase in the expression of GPNMB-expressing macrophages and CD44-positive cells in PH. Further studies are needed to identify the involvement of these cells in the pathogenesis of PH. Previous reports have suggested that CD44-positive cells express αSMA in PH model mice [8], and our data in humans and rats confirmed the expression of CD44 around the pulmonary arteries. The human lung tissue analysis confirmed increased GPNMB or CD44 expression in patients with PH, demonstrate the translational importance. Although we observed correlations between the changes in gut microbiota and the decrease in GPNMB+ macrophages and CD44 positive cells, further studies are needed to explore the more specific mechanisms of their interaction, and to validate these findings in human PH patients with more complex disease heterogeneity.

As a clinical treatment for PH, the greatest advantage of lactic acid bacteria probiotics is that they are highly safe and have rarely potential for conflict with other treatments or side effects. The limitations of this study include the fact that (1) the single probiotic strain tested; (2) the pathology of PH in the rat MCT model differs from the clinical findings in patients; (3) the lack of longitudinal microbiome/metabolome monitoring; and that (4) the rats were kept in an Specific pathogen Free (SPF) environment, which is not optimal for analyzing the gut microbiome. Probio-M9 needs to be tested in other PH models and its potential clinical application, dosage, duration of treatment, and combination therapy should be investigated.

## 5. Conclusions

These findings suggest a novel mechanism for the treatment of PH, in which the probiotic Probio-M9 regulates SCFA/NO/p-cresol metabolism in the gut microbiota and regulates GPNMB-positive macrophages and CD44-positive cells around the pulmonary arteries. Detailed studies of the mechanisms of gut-lung axis communication may provide a more comprehensive perspective for understanding the development and progression of PH. Probio-M9 may be investigated as a novel microbiome-targeted intervention for PH.

## Figures and Tables

**Figure 1 nutrients-17-02927-f001:**
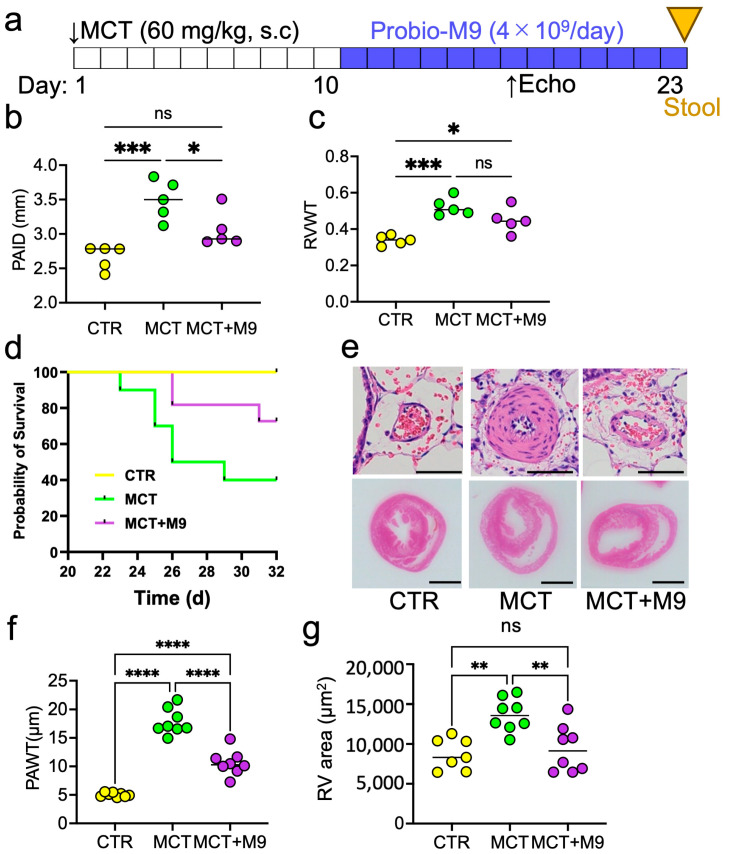
Therapeutic effects of Probio-M9 in monocrotaline-induced PH model rats. (**a**) Experimental protocol, the arrows showing the timing of pharmacological intervention and evaluation. (**b**,**c**) Summary of hemodynamic analysis by echocardiography on day 18. Pulmonary artery internal diameter (PAID) and right ventricular wall thickness (RVWT) are summarized (*n* = 5). (**d**) Survival curves of control rats (CTR) and monocrotaline-treated rats without (MCT) and with (MCT + M9) treatment with Probio-M9 (*n* = 10). (**e**) Representative images of HE staining of the rat pulmonary artery and heart. (**f**,**g**) Pulmonary artery wall thickneth (PAWT) and right ventricular wall area (RV area) in HE staining data are summarized (*n* = 8). Scale bar: 50 μm in upper panel, 0.5 cm in lower panel. * *p* < 0.05, ** *p* < 0.01, *** *p* < 0.001, **** *p* < 0.0001, ns, not significant.

**Figure 2 nutrients-17-02927-f002:**
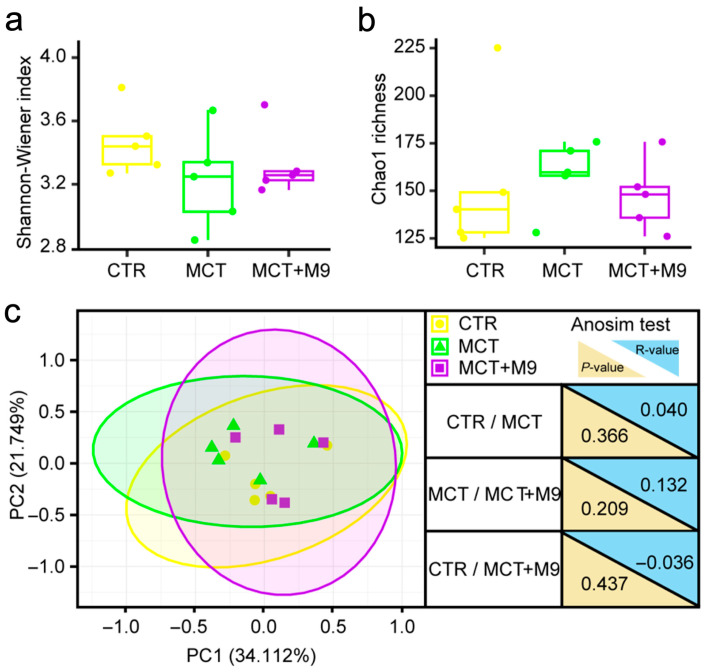
Microbial diversity after Probio-M9 supplementation in rats. (**a**,**b**) Shannon–Wiener index and Chao 1 richness of the fecal microbiome after Probio-M9 supplementation in rats. Statistical differences in the Shannon–Wiener index and Chao 1 richness between the groups were evaluated using the unpaired Wilcoxon test. Boxplot elements: center line, median; box limits, upper and lower quartiles; whiskers, 1.5× interquartile range; points, the value corresponding to the samples. (**c**) Principal coordinate analysis (PCoA) score plots after Probio-M9 supplementation in rats. Samples of each group are represented by different colors. *p*-value and R-value of the analysis of similarities (Anosim test, 999 permutations). (*n* = 5).

**Figure 3 nutrients-17-02927-f003:**
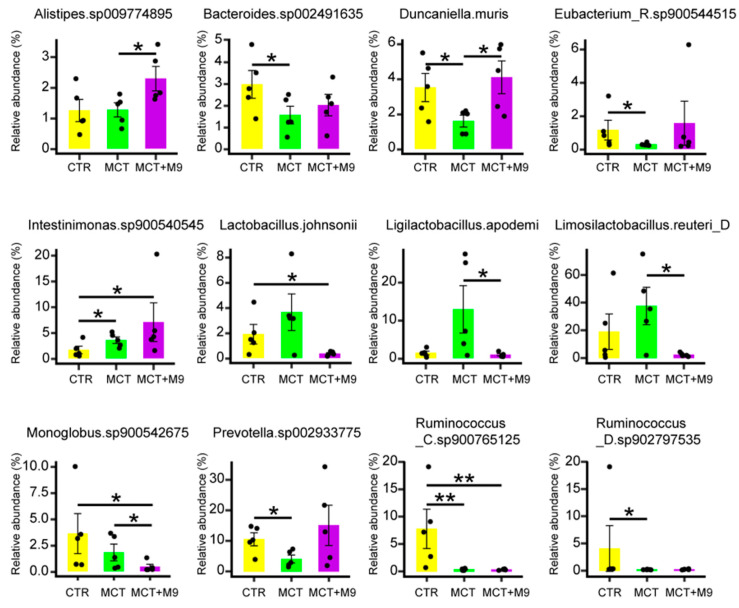
Profiles of significantly different species and gut metabolic modules among groups after Probio-M9 supplementation. Comparison of reads per kilobase million (RPKM) of different species between different groups after Probio-M9 supplementation. Significant differences in specific species between the two groups were evaluated using an unpaired Wilcoxon test (*n* = 5, * *p* < 0.05, ** *p* < 0.01).

**Figure 4 nutrients-17-02927-f004:**
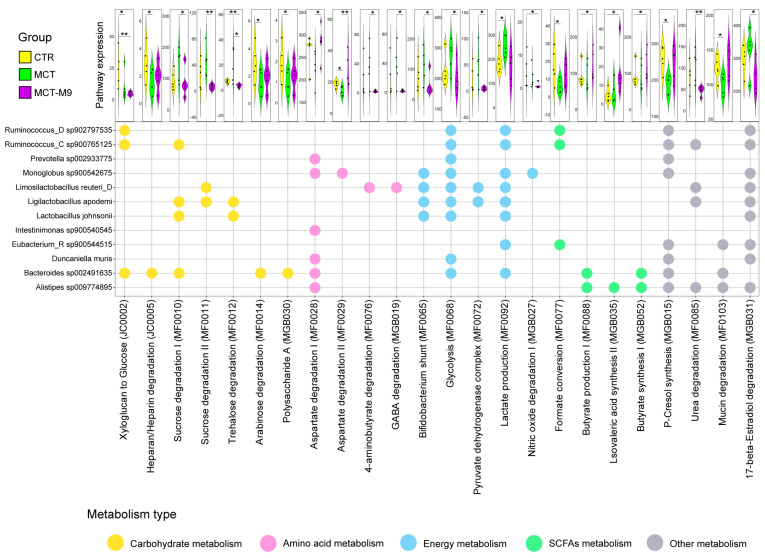
Differences in metabolic modules among the three groups. Top panel: Violin plot comparing the abundance of different gut metabolic modules between different groups after Probio-M9 supplementation. Significant differences in specific species between the two groups were evaluated using an unpaired Wilcoxon test. Lower panel: Significantly different metabolic modules across significantly different species (*n* = 5, * *p* < 0.05, ** *p* < 0.01).

**Figure 5 nutrients-17-02927-f005:**
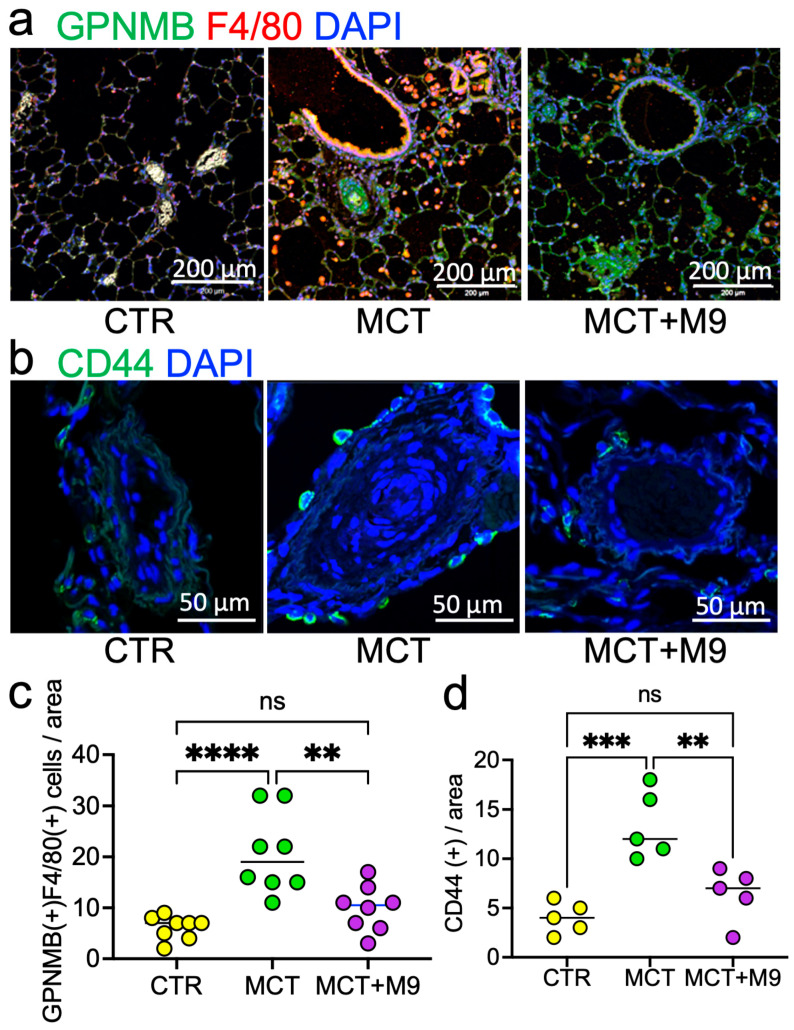
GPNMB and CD44 expression in MCT PH rat’s lung. (**a**) Representative images of immunofluorescence staining for GPNMB and F4/80 in rat lungs. (**b**) Representative images of immunofluorescence staining for CD44 around the PA in rat lungs. (**c**) Numbers of GPNMB and F4/80 double-positive cells in the area (0.36 mm^2^) around the PA are summarized (*n* = 8). (**d**) Number of CD44-positive cells in the area (0.04 mm^2^) around the PA (*n* = 5). ** *p* < 0.01, *** *p* < 0.001 or **** *p* < 0.0001, ns, not significant.

**Figure 6 nutrients-17-02927-f006:**
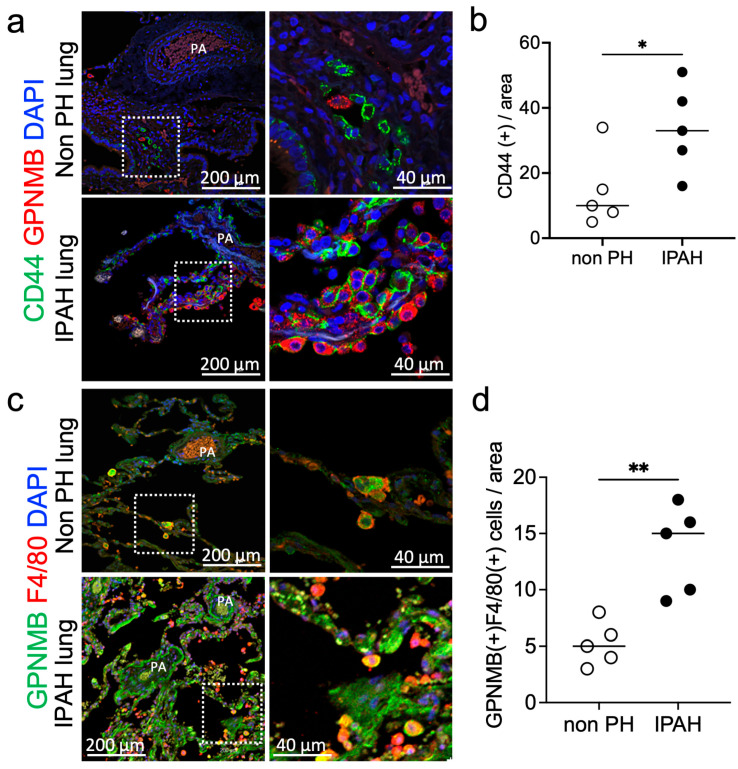
GPNMB and CD44 expression in PH patient’s lung. (**a**) Representative images of immunofluorescence staining for GPNMB and CD44 in the lungs of non-PH and PH patients. (**b**) Number of CD44-positive cells in the area (0.36 mm^2^) around the PA (*n* = 8). (**c**) Representative images of immunofluorescence staining for GPNMB and F4/80 around the PA in non-PH and PH patients. (**d**) The number of GPNMB and F4/80 double-positive cells in the area (0.36 mm^2^) around the PA (*n* = 5). * *p* < 0.05 or ** *p* < 0.01.

**Table 1 nutrients-17-02927-t001:** List of antibodies used in the present study.

Antibody	Host Species	Dilution	Purpose	Product Information
CD44	Rabbit	1/1000	IH (rat, human)	(Abcam Cat# ab157107, Cambridge, UK)
F4/80	Rabbit	1/1000	IH (rat)	(Cell Signaling Cat# 70076, Danvers, MA, USA)
F4/80	Rabbit	1/1000	IH (human)	(DLdevelop Cat# DL98552A, Kelowna, BC, Canada)
GPMNB	Mouse	1/1000	IH (rat, human)	(Proteintech Cat# 66926-1, Rosemont, IL, USA)

**Table 2 nutrients-17-02927-t002:** Demographic information of patients with and without (Non-PH) pulmonary hypertension, whose samples were used for immunostaining.

	Sample Number	Age	Sex	Disease	Sampling Methods
Lung tissue	PH1	11	F	HPAH	biopsies
	PH2	23	F	IPAH	biopsies
	PH3	10	M	IPAH	biopsies
	PH4	25	M	IPAH	biopsies
	PH5	27	F	IPAH	biopsies
	NonPH1	14	M	Pneumothorax	biopsies
	NonPH2	22	M	Pneumothorax	biopsies
	NonPH3	27	M	Pneumothorax	biopsies
	NonPH4	16	M	Pneumothorax	biopsies
	NonPH5	22	M	Pneumothorax	biopsies

PH: Pulmonary Hypertension; F: Female; M: Male; HPAH: Heritable Pulmonary Arterial Hypertension; IPAH: Idiopathic Pulmonary Arterial Hypertension.

## Data Availability

All Data will be made available on request. The sequence dataset was deposited in the National Genomics Data Center (CNCB, https://ngdc.cncb.ac.cn/) Genome Sequence Archive (GSA: CRA022137), and publicly accessible at https://ngdc.cncb.ac.cn/gsa.

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
