# Peer review of "Lacticaseibacillus rhamnosus* Probio-M9 Alters the Gut Microbiota and Mitigates Pulmonary Hypertension in a Rat Model"

_nutrients, 2025, doi:10.3390/nu17182927_

Round 1
Reviewer 1 Report
Comments and Suggestions for Authors
Thank you for submitting the manuscript "Breast Milk-Derived Lacticaseibacillus rhamnosus Probio-M9 Alters the Gut Microbiota and Mitigates Pulmonary Hypertension in a Rat Model" to Nutrients. Although the experimental study is interesting, many corrections to the text must be made before it can be considered for publication:
- The title states that the probiotic is derived from breast milk, but if it is a probiotic that already has a commercial name, I don't think this information is necessary in the title.
- Line #26: It is necessary to define the abbreviation the first time it appears. If possible, avoid it in the abstract.
- Line #27: Which ones?
- Line #30: Comparison in the abstract? Not acceptable. The abstract is a presentation of this research.
- I suggest restructuring the introduction with the manuscript's theme in mind. Does the disease studied affect many people? Provide a background to the text. - Table 1: Why is the host species a rabbit and the species used in the study a rat? Is this common?
- The materials and methods section needs to include a section about the material (probiotic) used. Was it isolated at this time? What were the isolation conditions? What were the dietary inclusion conditions? Was the probiotic previously sequenced? What was the animals' diet like?
- The authors need to be careful with the statements made throughout the work, as this study only demonstrates results in animal models, and other models should still be tested. It is important to include a paragraph about the limitations at the end of the discussion to avoid making the work merely speculative.
Author Response
Reviewer 1
Thank you for submitting the manuscript "Breast Milk-Derived Lacticaseibacillus rhamnosus Probio-M9 Alters the Gut Microbiota and Mitigates Pulmonary Hypertension in a Rat Model" to Nutrients. Although the experimental study is interesting, many corrections to the text must be made before it can be considered for publication:
Response:
Thank you very much for carefully review of our manuscript. You have provided valuable insights for enhancing the quality of our manuscript. Please find a detailed point-by-point reply to each comment below. Please note that the changes we made according to reviewer comment are highlighted in yellow in the revised manuscript.
- The title states that the probiotic is derived from breast milk, but if it is a probiotic that already has a commercial name, I don't think this information is necessary in the title.
Response:
Thank you very much for the valuable comment. We have removed the [Breast Milk-Derived] part from the title.
- Line #26: It is necessary to define the abbreviation the first time it appears. If possible, avoid it in the abstract.
Response:
Thank you very much for your important comment, we define the abbreviation (MCT) the first time it appears in line 24.
- Line #27: Which ones?
Response:
Thank you very much. We added detailed microbiota to the revised manuscript as below.
Revised abstract: Specifically, Alistipes sp009774895 and Duncaniella muris populations increased, whereas Limosilactobacillus reuteri_D, Ligilactobacillus apodeme and Monoglobus sp900542675 decreased compared to those in the MCT group.
- Line #30: Comparison in the abstract? Not acceptable. The abstract is a presentation of this research.
Response:
Thank you very much. We changed the explanation as below.
Revised abstract: In lung tissue from PH patients, more GPNMB-positive macrophages were found than non-PH lungs, and an increase in CD44-positive cells was confirmed in the vicinity of GPNMB.
- I suggest restructuring the introduction with the manuscript's theme in mind. Does the disease studied affect many people? Provide a background to the text. - Table 1: Why is the host species a rabbit and the species used in the study a rat? Is this common?
Response:
Thank you very much. We added patients’ number of PH in the revised introduction. Table 1: If the host species of the primary antibody is rabbit, it means that an anti-rabbit antibody is used as the secondary antibody. There are no revisions to this comment.
Revised introduction: Referring to European data, PH was estimated at 5.8 cases per million adults annually, while its prevalence ranges from 47.6 to 54.7 cases per million adults.
- The materials and methods section needs to include a section about the material (probiotic) used. Was it isolated at this time? What were the isolation conditions? What were the dietary inclusion conditions? Was the probiotic previously sequenced? What was the animals' diet like?
Response:
Thank you very much for your important comment, the information about the isolation and characterization of Probio-M9 was described in reference 4. Probio-M9 was mixed in drinking water.
Revised method: Probio-M9 (4 x 109 CFU/day) was administered orally in drinking water starting 10 days after monocrotaline injection.
- The authors need to be careful with the statements made throughout the work, as this study only demonstrates results in animal models, and other models should still be tested. It is important to include a paragraph about the limitations at the end of the discussion to avoid making the work merely speculative.
Response:
Thank you very much for the suggestion. We revised more information from other article to the introduction about your suggestion.
Revised introduction: The single-cell RNA sequencing on NCBI Gene Expression Omnibus databases, analyzed lung tissue samples across healthy controls and Pulmonary arterial hypertension (PAH) patients to find the significant changes and potential functions of myeloid cell subsets in PAH patients and especially focused on GPNMB+ macrophages [8].
Revised introduction: The localization of GPNMB+ macrophages and CD44+ /αSMA+ cells have been identified in the mice PH model [8].
Reviewer 2 Report
Comments and Suggestions for Authors
This manuscript presents the results of an experimental study which evaluated the effects of the colostrum-derived bacterium Lacticaseibacillus rhamnosus Probio-M9 on the composition of the gut microbiota and on pulmonary hypertension in a rat model induced by monocrotaline. The authors report that Probio-M9 supplementation attenuated cardiovascular remodelling, reduced mortality and altered the intestinal microbiota profile. It also decreased the counts of GPNMB⁺ macrophages and CD44⁺ cells in lung tissue. This study addresses the emerging and clinically relevant research area of the gut–lung axis in PH and explores the potential of probiotics as a novel therapeutic approach.
I have made some suggestions on how you could improve your work. You don't have to agree with them or rewrite your work in the same way. They are simply intended to help you see things from a different perspective.
1) In the 'Introduction' section, please, explicitly state what gap in the literature this work fills. For example: 'Although the role of gut microbiota in PH has been explored, no studies to date have examined the effects of breast milk–derived Lactobacillus rhamnosus Probio-M9 on both gut microbiota composition and GPNMB/CD44 expression in PH.'
2) Please, briefly explain why targeting GPNMB⁺ macrophages and CD44⁺ cells could be a promising therapeutic strategy for PH. If possible, refer to any existing human data that links these markers to PH prognosis.
3) In the 'Introduction' section, please, include a brief summary of existing evidence linking the modulation of gut microbiota (through probiotics or prebiotics) to improved cardiovascular or pulmonary outcomes.
4) Please, briefly discuss how microbiota-driven changes might affect pulmonary vascular remodelling mechanistically (e.g. via systemic inflammation, metabolite production such as SCFAs, or immune cell trafficking).
5) Conclude the introduction with a concise hypothesis and objectives, for example: 'We hypothesised that Probio-M9 supplementation would remodel the gut microbiota and modulate immune responses in the lungs, thereby reducing vascular remodelling and improving outcomes in a monocrotaline-induced pulmonary hypertension (PH) rat model.'
6) In section 2.1 Monocrotaline (MCT)-induced PH model rats, please, include the number of animals per group for all experiments, not just some endpoints. Also, specify whether randomisation and blinding were applied. Provide the manufacturer details for monocrotaline.
7) In section 2.2 ' Echocardiography in rats' section, indicate the specific ultrasound probe model and frequency used, and clarify whether the operator performing the measurements was blinded.
8) In the '2.4. Clinical samples for histological evaluations' section, state the inclusion and exclusion criteria, and provide the ethical approval reference numbers.
9) Also, provide the software versions for all bioinformatic tools and explain the rationale behind certain parameter thresholds (e.g. completeness ≥80%, contamination ≤5%).
10) In the '2.9. Statistical Analyses' section, specify whether adjustments for multiple testing were applied, and justify the sample size statistically or by power analysis.
11) While some sections of the 'Results' state n values (e.g. for immunofluorescence), others (e.g. microbiota sequencing and metabolic module analysis) do not clearly specify how many animals were used per group. Sample sizes should be explicitly stated in each subsection to improve reproducibility.
12) In section 3.1 (survival/pathology), significant effects are described, but hazard ratios, survival percentages and mean ± SD changes for PA wall thickness, RVWT and PAID are not provided. Including these would provide a clearer indication of the magnitude of the effect.
13) In section 3.4, stating the fold change or percentage reduction in GPNMB⁺ macrophages and CD44⁺ cells would make the findings more impactful.
14) While it is good that specific species are mentioned in 3.2, the differences are only described qualitatively as 'more' or 'less'. Including relative abundance values or percentage changes would strengthen the evidence and enable the data to be interpreted without relying solely on figures.
15) Currently, sections 3.2 and 3.3 are separate. Adding a short linking sentence at the end of section 3.2 (e.g. 'We next assessed whether these compositional shifts were associated with functional metabolic changes...') would improve the flow.
16) In section 3.4, insert a clear transition sentence when moving from the rat data to the human patient results to explain why human lung samples were analysed and how they validate the animal findings.
17) Clearly state what is new compared to previous studies on probiotics or the gut microbiota in PH. Highlight whether Probio-M9 has previously been tested in a PH model or if modulation of Alistipes sp. in PH has been demonstrated. A short sentence in the opening paragraph of the 'Discussion' section could emphasise the novelty.
18) While the discussion acknowledges the stable diversity of the MCT model, it does not elaborate on the possible methodological limitations, such as sequencing depth, sampling time points, cage effects and diet standardisation. Credibility would be strengthened by noting the limitations of the rat PH model in translating to human PH.
19) Although Alistipes sp. is emphasised, other taxa with potential roles (e.g. Monoglobus sp.) are mentioned briefly. It would be useful to discuss how these species interact within the microbial network, and whether modulation of these species is secondary to changes in SCFAs or independent.
20) Currently, the discussion mostly focuses on local gut–lung immune interactions. Including more information on how gut microbiota-derived metabolites affect vascular remodelling, oxidative stress and right ventricular function could establish a stronger link between the findings and PH pathophysiology.
21) Please, consider the clinical implications, too. Could Probio-M9 or other SCFA-promoting probiotics be used as supportive therapy for human PH? Are there any safety considerations or potential contraindications? Could dietary interventions synergise with Probio-M9?
22) In the 'Discussion' section, some sentences reiterate the same point about SCFA-mediated immunomodulation. These could be condensed to avoid redundancy, allowing more space for broader discussion points.
23) Please, add a concluding synthesis paragraph summarising the main findings, the proposed mechanism (microbiota shift → SCFA/NO/p-cresol modulation → reduced macrophage-driven inflammation → alleviation of PH) and future research directions (e.g. human trials, long-term probiotic supplementation and combination therapies).
24) In the 'Conclusions' section, clearly state whether this is the first study to link Probio-M9 with PH improvement via the gut–lung axis. Highlight the novelty of identifying Alistipes sp. and metabolic pathways (SCFAs, NO and p-cresol) as potential mediators.
25) Briefly discuss how these preclinical findings could inform future probiotic-based interventions in human PH patients. Consider possible synergies with existing therapies or dietary interventions.
26) Restate the hypothesised mechanism in a single, impactful sentence (e.g. 'Our results suggest that modulation of specific gut microbial taxa and their metabolites can attenuate vascular remodelling in PH through immunomodulation of lung macrophages.').
27) Currently, the 'Conclusions' section only mentions general limitations. Consider specifying the following: 1) the single probiotic strain tested; 2) the use of the MCT-induced PH model (which may not capture all aspects of human PH pathophysiology); 3) the lack of longitudinal microbiome/metabolome monitoring.
28) Please, add a concise statement on future directions. (testing in other PH models, clinical feasibility studies, identifying optimal dosing and treatment duration, and exploring combination therapy potential).
29) Rather than ending with a general statement about the gut–lung axis, conclude with a forward-looking message that clearly signals relevance for future PH therapy development.
30) In the 'Abstract', clearly state why this study is unique. Is this the first time that Probio-M9 has been tested in pulmonary hypertension (PH)? Is the GPNMB/CD44 mechanism novel in the context of the gut–lung axis? Currently, the background information is too general and does not emphasise the knowledge gap.
31) To provide context, include the number of animals per group or the study duration. Briefly note that monocrotaline was used to induce PH. Where possible, include specific effect sizes (e.g. percentage reduction in mortality, changes in pulmonary artery wall thickness, or shifts in the relative abundance of key bacterial taxa). This would make the abstract more data-rich and credible.
32) Instead of saying that 'certain bacterial populations increased, whereas others decreased', specify which key taxa were altered (e.g. Alistipes sp. increased and Monoglobus sp. decreased). This would make a better connection with the discussion.
33) Specify that human lung tissue analysis was used to confirm increased GPNMB/CD44 expression in patients with PH, to demonstrate the study's translational value.
34) Rather than ending with a descriptive sentence, finish with a more impactful statement. Highlight the potential therapeutic implications, for example, '...suggesting that Probio-M9 could be explored as a novel, microbiome-targeted intervention for PH'.
This manuscript contains valuable findings, but improving its clarity, the depth of its discussion and its methodological detail would further enhance its impact.
Author Response
Reviewer 2
This manuscript presents the results of an experimental study which evaluated the effects of the colostrum-derived bacterium Lacticaseibacillus rhamnosus Probio-M9 on the composition of the gut microbiota and on pulmonary hypertension in a rat model induced by monocrotaline. The authors report that Probio-M9 supplementation attenuated cardiovascular remodelling, reduced mortality and altered the intestinal microbiota profile. It also decreased the counts of GPNMB⁺ macrophages and CD44⁺ cells in lung tissue. This study addresses the emerging and clinically relevant research area of the gut–lung axis in PH and explores the potential of probiotics as a novel therapeutic approach.
I have made some suggestions on how you could improve your work. You don't have to agree with them or rewrite your work in the same way. They are simply intended to help you see things from a different perspective.
Response:
We deeply appreciate the time and expertise you have dedicated to reviewing our work. Your detailed comments reflect a profound understanding of our research. We have carefully addressed each of your comments and made corresponding revisions to improve the clarity and scientific rigor of our work. Please note that the changes we made according to reviewer comment are highlighted in yellow in the revised manuscript.
1) In the 'Introduction' section, please, explicitly state what gap in the literature this work fills. For example: 'Although the role of gut microbiota in PH has been explored, no studies to date have examined the effects of breast milk–derived Lactobacillus rhamnosus Probio-M9 on both gut microbiota composition and GPNMB/CD44 expression in PH.'
Response:
Thank you very much for the important suggestion, we revised the manuscript according to your suggestion as below.
Revised introduction: Although the role of gut microbiota in PH has been explored, no studies to date have examined the effects of Probio-M9 on both gut microbiota composition and GPNMB / CD44 expression in PH.
2) Please, briefly explain why targeting GPNMB⁺ macrophages and CD44⁺ cells could be a promising therapeutic strategy for PH. If possible, refer to any existing human data that links these markers to PH prognosis.
Response:
Thank you very much for the important suggestion. We added the important information to the revised manuscript.
Revised introduction: The single-cell RNA sequencing on NCBI Gene Expression Omnibus databases, analyzed lung tissue samples across healthy controls and Pulmonary arterial hypertension (PAH) patients to find the significant changes and potential functions of myeloid cell subsets in PAH patients and especially focused on GPNMB+ macrophages [8].
Revised introduction: The localization of GPNMB+ macrophages and CD44+ /αSMA+ cells have been identified in the mice PH model [8].
3) In the 'Introduction' section, please, include a brief summary of existing evidence linking the modulation of gut microbiota (through probiotics or prebiotics) to improved cardiovascular or pulmonary outcomes.
4) Please, briefly discuss how microbiota-driven changes might affect pulmonary vascular remodelling mechanistically (e.g. via systemic inflammation, metabolite production such as SCFAs, or immune cell trafficking).
Response to 3) and 4):
Thank you very much for the very important suggestion. We revised introduction part as below.
Revised introduction: It has been suggested that adjusting the gut flora may also contribute to lung health and that bacterial supplements can ameliorate or induce PH, affect pulmonary vascular re-modeling mechanistically (e.g. via systemic inflammation, metabolite production such as SCFAs, or immune cell trafficking) [19-22].
5) Conclude the introduction with a concise hypothesis and objectives, for example: 'We hypothesised that Probio-M9 supplementation would remodel the gut microbiota and modulate immune responses in the lungs, thereby reducing vascular remodelling and improving outcomes in a monocrotaline-induced pulmonary hypertension (PH) rat model.'
Response:
Thank you very much for suggestion.
Revised introduction: We hypothesized that Probio-M9 supplementation would remodel the gut microbiota and modulate immune responses in the lungs, thereby reducing vascular remodeling and improving outcomes in a monocrotaline (MCT)-induced PH rat model.
6) In section 2.1 Monocrotaline (MCT)-induced PH model rats, please, include the number of animals per group for all experiments, not just some endpoints. Also, specify whether randomisation and blinding were applied. Provide the manufacturer details for monocrotaline.
Response:
Thank you very much for your helpful comment, we added all information you mentioned.
Revised method: Eight-week-old male Sprague-Dawley rats received a subcutaneous injection of mono-crotaline (Sigma-Aldrich Chemical Co., St. Louis, MO) (60 mg/kg) on day 1 (Figure 1a). Probio-M9 (4 x 109 CFU/day) was administered orally in drinking water starting 10 days after monocrotaline injection. Two sets of experiments were conducted: one for the evaluation of cardiac function on day 18 and survival until day 32 (n=10), and the other for the collection of tissues and stools for evaluation on day 23 (n=8). Randomization and blinding were applied for all experiments.
7) In section 2.2 ' Echocardiography in rats' section, indicate the specific ultrasound probe model and frequency used, and clarify whether the operator performing the measurements was blinded.
Response:
Thank you very much for the helpful comment, we added probe information and blinding policy to the revised manuscript.
Revised method: Echocardiography was performed performed in a blinded manner using a LOGIQ ul-trasound machine (GE Healthcare) and equipped with a 5–11.5 MHz multifrequency probe.
8) In the '2.4. Clinical samples for histological evaluations' section, state the inclusion and exclusion criteria, and provide the ethical approval reference numbers.
Response:
Thank you very much for suggestion. Please check the ethical approval reference numbers were listed in [Institutional Review Board Statement] section at the end of the manuscript. We did not make any change about this point.
9) Also, provide the software versions for all bioinformatic tools and explain the rationale behind certain parameter thresholds (e.g. completeness ≥80%, contamination ≤5%).
Response: Thank you for your comment. We have included the software versions for all bioinformatic tools and explain the rationale behind certain parameter thresholds. Please see section 2.6-2.9 in the revised manuscript. Thank you again for your question.
10) In the '2.9. Statistical Analyses' section, specify whether adjustments for multiple testing were applied, and justify the sample size statistically or by power analysis.
Response:
Thank you very much for the important point, we did multiple testing, and justified the sample size statistically. We added number of rats and the statistical analysis of animal study in the revised method section.
Revised method: The n indicates the number of independent experiments or the number of animals. Differences in numerical variables among groups were evaluated using analysis of variance, followed by the Tukey–Kramer test for multiple comparisons. Statistical sig-nificance was set at P < 0·05. All statistical analyses were performed using Prism-GraphPad ver9.5.1 (GraphPad Software, San Diego, CA, USA).
11) While some sections of the 'Results' state n values (e.g. for immunofluorescence), others (e.g. microbiota sequencing and metabolic module analysis) do not clearly specify how many animals were used per group. Sample sizes should be explicitly stated in each subsection to improve reproducibility.
Response:
Thank you very much for suggestion. We added n=5 to revised figure 2~4.
12) In section 3.1 (survival/pathology), significant effects are described, but hazard ratios, survival percentages and mean ± SD changes for PA wall thickness, RVWT and PAID are not provided. Including these would provide a clearer indication of the magnitude of the effect.
Response:
Thank you very much for important suggestion. In the Results section, there was a citation error in the figure numbers and order, so this has been corrected to cite all figures in the revised manuscript.
Revised Result: Echocardiography on day18 showed that Probio-M9 treatment significantly improved the decreased pulmonary artery internal diameter (PAID), and right ventricular wall thickness (RVWT) compared with the MCT group (Figure 1b, c). Histopathology on day 23 showed that Probio-M9 treatment significantly decreased the PA wall and RV thickness compared to the MCT group (Figure 1e~g).
13) In section 3.4, stating the fold change or percentage reduction in GPNMB⁺ macrophages and CD44⁺ cells would make the findings more impactful.
Response:
Thank you very much for suggestion. We added median value in the results section.
Revised Result: We found that the number of CD44 positive cells (CTR 4 vs MCT 12 vs MCT+M9 7 in median value)and GPNMB-positive macrophages (CTR 7 vs MCT 19 vs MCT+M9 10.5 in median value) in the PA region was significantly increased in the MCT group compared to the CTR group, whereas Probio-M9 treatment significantly reduced it (Figure 5a–d).
Revised Result: We also examined GPNMB and CD44 expression in the lungs of patients with and without PH and found that the number of CD44 positive cells (nonPH 12 vs PH 17 in median value) and GPNMB-positive macrophages (nonPH 5 vs PH 15 in median value) in the PA region was significantly higher in the PH group than in the non-PH group (Figure 6).
14) While it is good that specific species are mentioned in 3.2, the differences are only described qualitatively as 'more' or 'less'. Including relative abundance values or percentage changes would strengthen the evidence and enable the data to be interpreted without relying solely on figures.
Response:
Thank you very much. The results of metagenomic analysis showed many changes in specific species, and it is easier to understand visually if the changes are shown in figure rather than as values. We did not make any change about this point.
15) Currently, sections 3.2 and 3.3 are separate. Adding a short linking sentence at the end of section 3.2 (e.g. 'We next assessed whether these compositional shifts were associated with functional metabolic changes...') would improve the flow.
Response:
Thank you very much for the suggestion, we added [We next assessed whether these compositional shifts were associated with functional metabolic changes] to the revised manuscript.
16) In section 3.4, insert a clear transition sentence when moving from the rat data to the human patient results to explain why human lung samples were analysed and how they validate the animal findings.
Response:
Thank you very much for the good suggestion, we added [In the MCT rat lung, remodeling PAs were surrounded by numerous GPNMB+ or CD44+ cells. To evaluate the clinical evidences about these changes, we next analyzed the GPNMB+ or CD44+ cells localization and expression patterns clinical samples.] in the revised section 3.4.
17) Clearly state what is new compared to previous studies on probiotics or the gut microbiota in PH. Highlight whether Probio-M9 has previously been tested in a PH model or if modulation of Alistipes sp. in PH has been demonstrated. A short sentence in the opening paragraph of the 'Discussion' section could emphasise the novelty.
Response:
Thank you very much. We added [This study is the first to investigate the effects of Probio-M9 on both gut microbiota composition and GPNMB/CD44 expression in PH.] in the revised discussion. We have detailed description about the microbiota in the discussion section.
18) While the discussion acknowledges the stable diversity of the MCT model, it does not elaborate on the possible methodological limitations, such as sequencing depth, sampling time points, cage effects and diet standardisation. Credibility would be strengthened by noting the limitations of the rat PH model in translating to human PH.
Response:
Thank you very much for the comment, we added the limitation of this study.
Revised discussion: [The limitations of this study include the fact that 1) the single probiotic strain tested; 2) the pathology of PH in the rat MCT model differs from the clinical findings in patients; 3) the lack of longitudinal microbiome/metabolome monitoring; and that 4)the rats were kept in an Specific pathogen Free (SPF) environment, which is not optimal for analyzing the gut microbiome.]
19) Although Alistipes sp. is emphasised, other taxa with potential roles (e.g. Monoglobus sp.) are mentioned briefly. It would be useful to discuss how these species interact within the microbial network, and whether modulation of these species is secondary to changes in SCFAs or independent.
Response:
Thank you very much. We do not have enough evidence to proof how these species interact within the microbial network, and whether modulation of these species is secondary to changes in SCFAs or independent. So, we do not have any revision on this point.
20) Currently, the discussion mostly focuses on local gut–lung immune interactions. Including more information on how gut microbiota-derived metabolites affect vascular remodelling, oxidative stress and right ventricular function could establish a stronger link between the findings and PH pathophysiology.
Response:
Thank you very much. We mentioned about how gut microbiota-derived metabolites affect PH pathophysiology. We added [Gut microbiota-derived metabolites may influence vascular remodeling, oxidative stress, and right ventricular function, establishing a stronger link with PH pathophysiology.] in the revised discussion.
21) Please, consider the clinical implications, too. Could Probio-M9 or other SCFA-promoting probiotics be used as supportive therapy for human PH? Are there any safety considerations or potential contraindications? Could dietary interventions synergise with Probio-M9?
Response:
Thank you very much. We added [As a clinical treatment for PH, the greatest advantage of lactic acid bacteria probiotics is that they are highly safe and have rarely potential for conflict with other treatments or side effects.] in the revised discussion.
22) In the 'Discussion' section, some sentences reiterate the same point about SCFA-mediated immunomodulation. These could be condensed to avoid redundancy, allowing more space for broader discussion points.
Response:
Thank you very much. In the Discussion section, there are several statements about SCFA-mediated immune modulation, each from a different perspective. There is no overlap, so this point remains unchanged.
23) Please, add a concluding synthesis paragraph summarising the main findings, the proposed mechanism (microbiota shift → SCFA/NO/p-cresol modulation → reduced macrophage-driven inflammation → alleviation of PH) and future research directions (e.g. human trials, long-term probiotic supplementation and combination therapies).
Response:
Thank you very much.
Revised conclusions: [These findings suggest a novel mechanism for the treatment of PH, in which the probiotic Probio-M9 regulates SCFA/NO/p-cresol metabolism in the gut microbiota and regulates GPNMB-positive macrophages and CD44-positive cells around the pulmonary arteries. Detailed studies of the mechanisms of gut-lung axis communication may provide a more comprehensive perspective for understanding the development and progression of PH. Probio-M9 may be investigated as a novel microbiome-targeted intervention for PH.]
24) In the 'Conclusions' section, clearly state whether this is the first study to link Probio-M9 with PH improvement via the gut–lung axis. Highlight the novelty of identifying Alistipes sp. and metabolic pathways (SCFAs, NO and p-cresol) as potential mediators.
Response:
Thank you very much.
Revised discussion: [This study is the first to investigate the effects of Probio-M9 on both gut microbiota composition and GPNMB/CD44 expression in PH.]
25) Briefly discuss how these preclinical findings could inform future probiotic-based interventions in human PH patients. Consider possible synergies with existing therapies or dietary interventions.
Response:
Thank you very much. Revised discussion: [As a clinical treatment for PH, the greatest advantage of lactic acid bacteria probiotics is that they are highly safe and have rarely potential for conflict with other treatments or side effects.]
26) Restate the hypothesised mechanism in a single, impactful sentence (e.g. 'Our results suggest that modulation of specific gut microbial taxa and their metabolites can attenuate vascular remodelling in PH through immunomodulation of lung macrophages.').
Response:
Thank you very much. We added this description to the revised manuscript.
Revised discussion: [ Our results suggest that modulation of specific gut microbial taxa and their metabolites can attenuate vascular remodeling in PH through immunomodulation of lung macrophages.]
27) Currently, the 'Conclusions' section only mentions general limitations. Consider specifying the following: 1) the single probiotic strain tested; 2) the use of the MCT-induced PH model (which may not capture all aspects of human PH pathophysiology); 3) the lack of longitudinal microbiome/metabolome monitoring.
Response:
Thank you very much for the important comment.
Revised discussion: [The limitations of this study include the fact that 1) the single probiotic strain tested; 2) the pathology of PH in the rat MCT model differs from the clinical findings in patients; 3) the lack of longitudinal microbiome/metabolome monitoring; and that 4) the rats were kept in an Specific pathogen Free (SPF) environment, which is not optimal for analyzing the gut microbiome.]
28) Please, add a concise statement on future directions. (testing in other PH models, clinical feasibility studies, identifying optimal dosing and treatment duration, and exploring combination therapy potential).
Response:
Thank you very much for advice, we added future directions in the revised manuscript.
Revised discussion: [Probio-M9 needs to be tested in other PH models and its potential clinical application, dosage, duration of treatment, and combination therapy should be investigated.]
29) Rather than ending with a general statement about the gut–lung axis, conclude with a forward-looking message that clearly signals relevance for future PH therapy development.
Response:
Thank you very much. We changed the last paragraph of the conclusion as below: [An in-depth study of the mechanism of gut-lung axis communication will provide a more comprehensive perspective for understanding the occurrence and development of PH, and may reveal new therapeutic targets for the PH via microbiome.]
30) In the 'Abstract', clearly state why this study is unique. Is this the first time that Probio-M9 has been tested in pulmonary hypertension (PH)? Is the GPNMB/CD44 mechanism novel in the context of the gut–lung axis? Currently, the background information is too general and does not emphasise the knowledge gap.
Response:
Thank you very much. We added information about your concerns.
Revised introduction: [The single-cell RNA sequencing on NCBI Gene Expression Omnibus databases, analyzed lung tissue samples across healthy controls and Pulmonary arterial hypertension (PAH) patients to find the significant changes and potential functions of myeloid cell subsets in PAH patients and especially focused on GPNMB+ macrophages [8].]
Revised discussion: [This study is the first to investigate the effects of Probio-M9 on both gut microbiota composition and GPNMB/CD44 expression in PH.]
31) To provide context, include the number of animals per group or the study duration. Briefly note that monocrotaline was used to induce PH. Where possible, include specific effect sizes (e.g. percentage reduction in mortality, changes in pulmonary artery wall thickness, or shifts in the relative abundance of key bacterial taxa). This would make the abstract more data-rich and credible.
Response:
Thank you very much. We added number of rats in the revised manuscript. The results describe the use of monocrotaline to induce PH and the effect (e.g., percent reduction in mortality, change in pulmonary artery wall thickness, change in relative abundance of key bacterial taxa).
32) Instead of saying that 'certain bacterial populations increased, whereas others decreased', specify which key taxa were altered (e.g. Alistipes sp. increased and Monoglobus sp. decreased). This would make a better connection with the discussion.
Response:
Thank you very much for important comment.
Revised abstract: [Specifically, Alistipes sp009774895 and Duncaniella muris populations increased, whereas Limosilactobacillus reuteri_D, Ligilactobacillus apodeme and Monoglobus sp900542675 decreased compared to those in the MCT group.]
33) Specify that human lung tissue analysis was used to confirm increased GPNMB/CD44 expression in patients with PH, to demonstrate the study's translational value.
Response:
Thank you very much. We added description as below:
Revised result section: [In the MCT rat lung, remodeling PAs were surrounded by numerous GPNMB+ or CD44+ cells. To evaluate the clinical evidences about these changes, we next analyzed the GPNMB+ or CD44+ cells localization and expression patterns clinical samples.]
Revised discussion: [We observed a significant increase in the expression of GPNMB-expressing macrophages and CD44-positive cells in PH. Further studies are needed to identify the involvement of these cells in the pathogenesis of PH. Previous reports have suggested that CD44-positive cells express αSMA in PH model mice, and our data in humans and rats confirmed the expression of CD44 around the pulmonary arteries. The human lung tissue analysis confirmed increased GPNMB or CD44 expression in patients with PH, demonstrate the translational importance.]
34) Rather than ending with a descriptive sentence, finish with a more impactful statement. Highlight the potential therapeutic implications, for example, '...suggesting that Probio-M9 could be explored as a novel, microbiome-targeted intervention for PH'.
Response:
Thank you very much. We added the description [Probio-M9 could be explored as a novel, microbiome-targeted intervention for PH.] in the revised conclusion.
This manuscript contains valuable findings, but improving its clarity, the depth of its discussion and its methodological detail would further enhance its impact.
Response:
Thank you very much for all detailed comments.
Reviewer 3 Report
Comments and Suggestions for Authors
Thank you for the opportunity to participate in the review of the manuscript titled “Breast Milk-Derived Lacticaseibacillus rhamnosus Probio-M9 Alters the Gut Microbiota and Mitigates Pulmonary Hypertension in a Rat Model.”
The manuscript describes the study's effect on the development of pulmonary hypertension (PH) and changes in the gut microbiota. The study was conducted in rats. The manuscript follows the typical structure of a research article. The introduction to the topic is interesting and relevant, and the cited literature is accurate. The authors did not present a clear study objective, but the manuscript creates a logical structure, which facilitates the reading. The description of the material should be supplemented according to the notes below. The methodology is unclear in some places. For example, the control and study groups should be clearly listed and described. A brief description of abbreviations is provided only with figures. A corresponding description is missing in the methodology section. The description of the results and their presentation are correct. All necessary abbreviations should be explained below the figures. The project raises no ethical concerns. Appropriate consents have been obtained for animal and tissue studies.
This article addresses the important topic of the impact of probiotic bacteria on the health and metabolism of rats, aligning with current trends in gut microbiota research.
In summary, the manuscript is well-written and suitable for Nutrients after minor revisions.
Detailed comments:
Line 21. Please add the aim of the work to the abstract.
Line 26. Please explain the abbreviation MCT group.
Line 37. Please change the keywords to words other than those in the manuscript title. This will improve searchability in the article database. Furthermore, please add more keywords.
Line 73. What was the purpose of this study?
Line 75. Please add a subsection entitled "Material" and describe the strain Lacticaseibacillus rhamnosus Probio-M9. From what was it isolated (this information is in the title)? Does it have an accession number (e.g., Genbank)? What are its properties?
Line 77. The NIH abbreviation should be explained.
Line 88. Besides Figure 1a, why are these graphs not in the Results section?
Line 99. Please provide the manufacturer of isoflurane.
Table 2. The age should be added (days). The necessary abbreviations should be explained below the table.
Line 170. Again, the abbreviation. We all know NCBI, but it still needs an explanation.
Figure 4. This figure should be enlarged and its quality improved to make it more readable.
Author Response
Reviewer 3
The manuscript describes the study's effect on the development of pulmonary hypertension (PH) and changes in the gut microbiota. The study was conducted in rats. The manuscript follows the typical structure of a research article. The introduction to the topic is interesting and relevant, and the cited literature is accurate. The authors did not present a clear study objective, but the manuscript creates a logical structure, which facilitates the reading. The description of the material should be supplemented according to the notes below. The methodology is unclear in some places. For example, the control and study groups should be clearly listed and described. A brief description of abbreviations is provided only with figures. A corresponding description is missing in the methodology section. The description of the results and their presentation are correct. All necessary abbreviations should be explained below the figures. The project raises no ethical concerns. Appropriate consents have been obtained for animal and tissue studies.
This article addresses the important topic of the impact of probiotic bacteria on the health and metabolism of rats, aligning with current trends in gut microbiota research.
In summary, the manuscript is well-written and suitable for Nutrients after minor revisions.
Response: Thank you for review our manuscript and provider valuable feedback, which has helped us improve the clarity and rigor of our manuscript. Please note that the changes we made according to reviewer comment are highlighted in yellow in the revised manuscript.
Detailed comments:
Line 21. Please add the aim of the work to the abstract.
Response:
Thank you very much for suggestion. We added the aim of the study in revised manuscript: [We investigated whether Probio-M9 supplementation could improve the pathology of PH.]
Line 26. Please explain the abbreviation MCT group.
Response:
Thank you very much for suggestion, we added the abbreviation of MCT in the revised abstract.
Line 37. Please change the keywords to words other than those in the manuscript title. This will improve searchability in the article database. Furthermore, please add more keywords.
Response:
Thank you very much, we added key words [pulmonary artery remodeling, macrophages, GPNMB, CD44] in the revised manuscript.
Line 73. What was the purpose of this study?
Response:
Thank you very much for the important suggestion. We added the purpose of the study in revised manuscript.
Revised introduction: Although the role of gut microbiota in PH has been explored, no studies to date have examined the effects of Probio-M9 on both gut microbiota composition and GPNMB / CD44 expression in PH. We hypothesized that Probio-M9 supplementation would remodel the gut microbiota and modulate immune responses in the lungs, thereby reducing vascular remodeling and improving outcomes in a monocrotaline (MCT)-induced PH rat model.
Line 75. Please add a subsection entitled "Material" and describe the strain Lacticaseibacillus rhamnosus Probio-M9. From what was it isolated (this information is in the title)? Does it have an accession number (e.g., Genbank)? What are its properties?
Response:
Thank you very much for your important comment, the information about the isolation and characterization of Probio-M9 was described in reference 4. We removed the "Breast Milk-Derived" description from the title. Probio-M9 does not have Genbank accession number. The properties are listed in the first paragraph of introduction.
Line 77. The NIH abbreviation should be explained.
Response:
Thank you very much. I added "National Institutes of Health" instead of "NIH."
Line 88. Besides Figure 1a, why are these graphs not in the Results section?
Response:
Thank you very much for important suggestion. In the Results section, there was a citation error in the figure numbers and order, so this has been corrected to cite all figures in the revised manuscript.
Line 99. Please provide the manufacturer of isoflurane.
Response:
Thank you very much. We added the manufacturer information (Fujifilm-wako, Osaka Japan) to the revised method section.
Table 2. The age should be added (days). The necessary abbreviations should be explained below the table.
Response:
Thank you very much for the suggestion. We have ages information of patients in Table 2. We added the abbreviations below the Tables.
Revised Abreviations: PH: Pulmonary Hypertension, F: Female, M: Male, HPAH: Heritable Pulmonary Arterial Hypertension, IPAH: Idiopathic Pulmonary Arterial Hypertension
Line 170. Again, the abbreviation. We all know NCBI, but it still needs an explanation.
Response:
Thank you very much, we added the abbreviation of NCBI: [National Center for Biotechnology Information] to the revised manuscript.
Figure 4. This figure should be enlarged and its quality improved to make it more readable.
Response:
Thank you very much. The higher resolution file of the Images is revised to the editorial office.